# Anti-Inflammatory Activities of Constituents from *Cinnamomum insularimontanum* Hayata Leaves and Their Mechanisms

**DOI:** 10.3390/plants11233252

**Published:** 2022-11-26

**Authors:** Chieh-Yin Chen, Pei-Chen Wu, Nai-Wen Tsao, Yen-Hsueh Tseng, Fang-Hua Chu, Sheng-Yang Wang

**Affiliations:** 1Experimental Forest, College of Bio-Resources and Agriculture, National Taiwan University, Nantou County 55750, Taiwan; 2Department of Forestry, National Chung Hsing University, Taichung 40227, Taiwan; 3Special Crop and Metabolome Discipline Cluster, Academy Circle Economy, National Chung Hsing University, Taichung 40227, Taiwan; 4Taiwan Forestry Research Institute, Taipei 10551, Taiwan; 5School of Forestry and Resource Conservation, National Taiwan University, Taipei 10617, Taiwan; 6Agricultural Biotechnology Research Institute, Academia Sinica, Taipei 11529, Taiwan

**Keywords:** *Cinnamomum insularimontanum*, anti-inflammatory, essential oil, isoburmanol, burmanol, AP-1

## Abstract

*Cinnamomum insularimontanum* is an endemic species of Taiwan. Although most Cinnamomum plants have significant biological activity, the bioactivity investment of *C. insularimontanum* is rare. Since inflammation plays an important role in many diseases, anti-inflammatory compounds can be developed into healthcare products. Therefore, we first conducted a study on the anti-inflammatory activity of *C. insularimontanum* leaves. First, we examined the antiinflammation activity of essential oil from *C. insularimontanum* leaves, and it revealed potent anti-inflammatory activity. A total of 23 volatile compounds were identified in *C. insularimontanum* leaves’ essential oil by using GC/MS analysis. Among them were 1,8-cineole (35.94%), α-eudesmol (6.17%), pinene (7.55%), sabinene (5.06%), and isobornyl acetate (4.81%). According to previous studies, 1,8-cineole might be an anti-inflammation principal compound of *C. insularimontanum* leaves. Next, the ethanolic extracts of *C. insularimontanum* leaves also exhibited good anti-inflammatory activity. Two bioactive compounds, isoburmanol (**F1**) and burmanol (**F2**), were isolated from the ethyl acetate soluble fraction by using the bioactivity-guided separation protocol and spectroscopic analysis. **F1** was obtained from *C. insularimontanum* for the first time, and **F2** was isolated for the first time from natural resources. Both **F1** and **F2** could inhibit the production of nitric oxide (NO), and the IC_50_ values were 14.0 μM and 43.8 μM, RAW 264.7 cells after induction of lipopolysaccharide. Furthermore, **F1** and **F2** also revealed significant inhabitation effects on iNOS and COX-2 protein expression. The anti-inflammation activity of **F1** and **F2** was different from the common pathway of inhibiting NF-κB. Both of them could inhibit the production of NO and PGE_2_ by directly inhibiting the AP-1 (c-Jun) protein and then inhibiting the downstream iNOS and COX-2. Although both **F1** and **F2** possessed significant anti-inflammatory activity, the activity of **F1** was better than **F2**. Through molecular docking simulation analysis, the results show that **F1** and **F2** interact with AP-1, inhibit the binding of AP-1 to DNA, and cause AP-1 to fail to transcribe the related factors of inflammation. The binding ability of AP-1 and **F1** was stronger than **F2**, and that is the reason why **F1** exhibited better activities in both downstream proteins and inflammatory cytokines. Based on the results obtained in this study, the essential oil and **F1** and **F2** isolated from *C. insularimontanum* leaves have good anti-inflammatory activities, and it is expected to be used as a reference for the development of medical care products in the future.

## 1. Introduction

Inflammation is at the basis of various diseases, such as cancer, diabetes, asthma, inflammatory bowel disease, multiple sclerosis, and Alzheimer’s disease, and can contribute to their progression [1]. In fact, it is a normal and healthy response in the human immune system; however, when the symptoms of inflammation are prolonged in the body, it begins to damage the normal cells and becomes harmful, creating a pro-inflammatory state [2]. According to the study by Wawrzyniak-Gramacka and his co-worker, they conclude that increased consumption of anti-inflammatory diet ingredients and physical activity sustained throughout life attenuates the progression of ageing, and chronic, sterile, low-grade inflammation (i.e., inflammaging) in elderly citizens. It is also a potential therapeutic strategy to counteract the pathophysiological effects of ageing [3].

Anti-inflammatory drugs can be mainly divided into two classes, i.e., glucocorticoids and non-steroidal anti-inflammatory drugs (NSAIDs). The mechanism of these anti-inflammation drugs is different. Briefly, glucocorticoids work by inhibiting prostaglandins and proteins involved in the inflammatory process, such as corticosteroids, which are also used in other indications to treat asthma and autoimmune inflammation. Non-steroidal anti-inflammatory drugs (NSAIDs) work by inhibiting cyclooxygenase and controlling moderate and mild pain and fever. An example of a non-steroidal drug is acetylsalicylic acid [4]. Nature is the original treasure source for medicines and health-promotion products [5]. Since ancient times, humans have used plants to treat diseases or use them as health care products. Now many famous drugs are developed from plant sources, such as aspirin [6] and paclitaxel [7].

*Cinnamomum insularimontanum* Hayata (Lauraceae), known as Taiwan cinnamon, is an endemic plant to Taiwan. It is distributed in the middle and low-altitude forests of the whole island. The main components of the essential oil of *C. insularimontanum* fruits are citral (35.89%), citronellal (24.64%), citronellol (16.78%), α-pinene (9.45%), β-pinene (4.30%), limonene (1.76%), and camphene (1.70%) [8]. It is known that the methanol extract of the *C. insularimontanum* stem has β-sitosterol, stigmasterol, coumarin, cinnamyl alcohol, cinnamic acid, p-hydroxybenzoic acid, kaempferol, and kaempferitrin [9]; the methanol extract of leaves has compounds such as obtusilactone A, isoobtusilactone A, (+)-sesamin, (+)-diasesamin, (+)-episesamin, β-sitostenone, β-sitosterol, cinnamic acid, and eugenol [10]. *C. insularimontanum* has been reported to have some biological activities. The fruit’s essential oil possessed potent in vitro and in vivo anti-inflammation activities [8]. Actinodaphnine isolated from the roots of *C. insularimontanum* has proven that it could induce apoptosis through increased nitric oxide, reactive oxygen species, and down-regulation of NF-kB signaling in human hepatoma Mahlavu cells [11]. Meanwhile, water extracts of *C. insularimontanum* possessed the anti-human herpes simplex virus 2 (HSV-2) efficacy with IC_50_ of 180 ± 30 μg/mL [12]. Although we had demonstrated the anti-inflammation activity of *C. insularimontanum* fruits, the bioactivity investigation for leaves is rare [8]. In this study, the essential oil was obtained from *C. insularimontanum* leaves, and its compositions and anti-inflammation activity were characterized and evaluated. On the other hand, the anti-inflammatory components of ethanolic extracts of *C. insularimontanum* leaves were also obtained following the bioactivity-guided fractionation procedure. According to our findings, extracts are expected to be used and compounds isolated from *C. insularimontanum* leaves as a reference for the development of medical care products in the future.

## 2. Results and Discussion

### 2.1. Chemical Composition Analysis and Its Anti-Inflammatory Activity of C. insularimontanum Leaves Essential Oil (CILE)

The yield of CILE by hydro-distillation was 2.1 mL/kg. The major chemical constituents of CILE were determined by GC–MS analysis. The relative contents (%) of major compounds in CILE are shown in Table 1. A total of 23 volatile compounds were identified in CILE, comprising 86.1%. Among them, 1,8-cineole (35.94%), α-eudesmol (6.17%), pinene (7.55%), sabinene (5.06%), and isobornyl acetate (4.81%) were the major components, which made up around 69.56% of the content of the CILE.

The MTT test was used to evaluate the cytotoxicity of the CILE on RAW 264.7 first. As shown in Figure 1, the cell viability was higher than 100% at the dosages of 3, 6, 12, 25, 50, and 100 μg/mL of CILE. It indicated no significant cytotoxicity of CILE on RAW 264.7 cells under the CILE concentration of 100 μg/mL. The nitric oxide (NO) inhibitory activity of leaves essential oil of *C. insularimontanum* (CILE) was evaluated by using an LPS-stimulated RAW 264.7 cell assay. As shown in Figure 1, the CILE revealed significant inhibitory activity on NO production in LPS-stimulated RAW 264.7 cells. RAW 264.7 cells treated with CILE at dosages of 3–100 μg/mL caused a dose-dependent NO inhibitory activity. The NO inhibition rate at the dosages of 3, 6, 12, 25, 50, and 100 μg/mL of CILE were 7.3%, 13.4%, 34.2%, 54.4%, 74.1%, and 91.4%, respectively, and the IC_50_ was 22.8 ± 2.9 μg/mL. The IC_50_ of the reference compound, curcumin, was 3.7 ± 0.1 μg/mL. Compared with the previous study, the essential oil of *C. insularimontanum* fruit (IC_50_ = 18.7 μg/mL) and Zingiber cassumunar essential oil (IC_50_ = 21.5 μg/mL) mL) was similar [8,13]. According to results obtained in this study, CILE revealed good anti-inflammatory activity.

### 2.2. Anti-Inflammatory Activity of Ethanolic Extract and Its Bioactive Compounds Isolation

The fresh leaves of *C. insularimontanum* were extracted with 70% ethanol (EtOH) to give crude extracts (Crude). The extract was dried and then partitioned with solvents of increasing polarity: *n*-hexane, ethyl acetate, and 1-butanol, successively. After removing solvents from the combined extractives, the *n*-Hex, EA, BuOH soluble fractions, and water-soluble fractions were obtained. The cytotoxicity and anti-inflammatory activity of Crude and its subtractions were evaluated by using an MTT assay and NO production in LPS-stimulated RAW 264.7 cell model (Figure 2). The EA soluble fraction presented a more potent anti-inflammatory activity than other fractions. Subsequently, the EA soluble fraction subjected to the silica gel column chromatography yielded ten sub-fractions (A–J). These sub-fractions were evaluated for their anti-inflammatory activity, and the results indicate that the F sub-fractions have the strongest anti-inflammatory activity (Table 2). The F sub-fraction was purified by normal phase HPLC to obtain two compounds (**F1** and **F2**). The structures of **F1** and **F2** were elucidated by NMR spectroscopy and mass spectrometry (Figure 3).

Compound **F1** was isolated as a white amorphous powder from sub-fraction F. The molecular formula C_17_H_16_O_4_ was determined by ESI-MS at *m/z* 283.0 [M-H]^-^ and ^13^C NMR data, indicating ten degrees of unsaturation. The NMR data are shown in Table 3. The ^1^H NMR spectrum displayed a signal for a methoxy group at *δ*_H_ 3.92 (3H, s); an olefinic proton at *δ*_H_ 6.13 (1H, t, *J* = 7.4 Hz); four methylene protons at *δ*_H_ 2.72 (1H, dd, *J* = 12.6, 6.3 Hz), *δ*_H_ 3.03 (1H, dd, *J* = 12.6, 8.2 Hz), *δ*_H_ 4.33 (1H, d, *J* = 13.0 Hz), *δ*_H_ 4.45 (1H, d, *J* = 13.0 Hz); five aromatic protons, including two para-oriented benzene protons at *δ*_H_ 7.03 (1H, s) and *δ*_H_ 7.18 (1H, s); and a set of ABX coupling protons at *δ*_H_ 6.65 (1H, d, *J* = 2.5 Hz), *δ*_H_ 6.70 (1H, dd, *J* = 8.3, 2.5 Hz), and *δ*_H_ 7.26 (1H, d, *J* = 8.3 Hz). The ^13^C NMR, DEPT, and HSQC experiments displayed 17 carbon signals, including two aromatic systems at *δ*_C_ 131.5 (C-1), 114.1 (C-2), 158.1 (C-3), 114.0 (C-4), 131.8 (C-4a), 129.6 (C-7a), 110.8 (C-8), 147.8 (C-9), 146.5 (C-10), 117.2 (C-11), 135.6 (C-11a), 145.1 (C-11b); two olefinic carbons at *δ*_C_ 128.0 (C-6) and 139.1 (C-7); two methylene carbons at *δ*_C_ 34.1 (C-5); and 66.4 (C-12), C-3, and C-12 were liked to oxygen, a methoxy carbon, at *δ*_C_ 56.6. Two benzene rings and an olefinic bond suggested nine degrees of unsaturation. Since the degrees of unsaturation is ten, indicating that **F1** had a tricyclic structure. Analysis of HSQC and HMBC spectra revealed H-1/C-3; H-2/C-3; H-8/C-10; H-11/C-10 suggesting the hydroxyl group was located at C-7 and C-10. The HMBC experiment showed H-12/C-6, C-7, C-7a, and H-5/C-6; C-7 indicated the hydroxymethyl group was attached to C-7 (Figure 4). The HMBC correlation between the methoxy group and C-9, and the NOESY correlation between H-12/H-8; 9-OMe, and H-8 confirmed the methoxy group linked to C-9, as shown in Figure 4. Based on the above evidence and comparison of spectroscopic (^1^H and ^13^C NMR) in the literature [14,15,16], compound **F1** was elucidated as 7-(hydroxymethyl)-9-methoxy-5*H*-dibenzo[a,c] [7] annulene 3,10-diol, which is the first time isolated from nature and named isoburmanol (Figure 3).

Compound **F2** was obtained as a white amorphous powder from sub-fraction F. The molecular formula was established to be C_17_H_16_O_4_ from ESI-MS at *m/z* 283.1 [M-H]^-^ and ^13^C NMR spectrum. The 1D and 2D NMR spectra were similar to that of **F1**, except for some differences, which were at C-1 (*δ*_H_ 7.32, *δ*_C_ 131.4), C-8 (*δ*_H_ 7.06, *δ*_C_ 114.1), and C-11 (*δ*_H_ 7.08, *δ*_C_ 113.7), as shown in Table 4. The HMBC signal displayed correlations of the methoxy group to C-10 (Figure 4). The NOESY spectrum showed correlations of H-8/H-12; 10-OMe/H-11 confirmed the location of 10-OMe. Based on this evidence and compared with the literature [17], compound **F2** was identified as burmanol.

### 2.3. Anti-Inflammatory Activity of Isoburmanol (***F1***) and Burmanol (***F2***)

#### 2.3.1. NO Production and PGE2 Synthesis Inhibitory Activity

The IC_50_ of **F1** and **F2** for the inhibition of NO production were 14.2 ± 3.6 μM and 44.7 ± 8.4 μM, respectively (Figure 5). In line with the Griess assay of NO production, the tested cells were healthy and viable at these concentrations of **F1** and **F2** (Figure 5). Next, the inhibition of PGE_2_ production was further detected. After induction with LPS, the production of PGE2 reached 16.1 μg/mL. In the group treated with **F1**, the production of PGE_2_ was less than 1 μg/mL at concentrations of 20, 40, and 60 μM, and almost no PGE_2_ was produced. In the **F2**-treated group, the amounts of PGE_2_ were 9.5, 3.0, 1.0, and 0.4 μg/mL at different concentrations of 20, 40, 60, and 80 μM, respectively (Figure 6). In the same assay, lucidone was used as a reference compound. It has been proved that lucidone has a good anti-inflammatory effect and can reduce the generation of PGE_2_ induced by LPS [18]. Lucidone also has the effect of reducing the generation of PGE_2_ in this experiment.

#### 2.3.2. Effects of **F1** and **F2** on iNOS Expression in LPS-Induced Macrophages

In order to confirm whether the inhibitory effect of **F1** and **F2** on NO production was regulated by iNOS, the protein expression of iNOS was analyzed by Western blotting. The results are shown in Figure 7. The iNOS protein was significantly increased after LPS induction, and **F1** (dosages = 20, 40, and 60 μM), **F2** (dosages = 40, 60, and 80 μM), and reference compound curcumin (15 μM) and lucidone (40 μM) treatment could significantly reduce the expression of iNOS protein with a dose-dependent manner. Especially, at 60 μM concentration **F1** treatment, no iNOS expression was observed, and it was similar to the reference compounds. The activity was even better than the effect of curcumin and lucidone.

#### 2.3.3. Effects of **F1** and **F2** on Upstream Transcription Factors NF-κB and AP-1 Expression in LPS-Induced Macrophages

Since **F1** and **F2** could reduce NO and PGE2 production by NO and PGE2 production, we further explored which upstream proteins were involved to regulate the signal pathway. First, the transcription factors NF-κB was examined. The results show that NF-κB p50 and p65 were translocated into the nucleus after induction by LPS, that is the increased protein expression in the nucleus and a decrease in the cytoplasm (Figure 8). Both **F1** (20, 40, and 60 μM) and **F2** (40, 60, and 80 μM) treatments, could not prevent NF-κB p50 and p65 c transfer into the nucleus. It revealed that the NF-κB pathway did not involve the anti-inflammatory mechanism of **F1** and **F2**.

Since **F1** and **F2** did not inhibit iNOS and COX-2 through the NF-κB pathway, we speculated that **F1** and **F2** might regulate iNOS and COX-2 expression via the AP-1 pathway. AP-1 is a dimer composed of c-Jun and c-Fos, which is a common transcription factor that regulates inflammation. It is affected by the MAPK family [19] and is a member of the LPS/TLR4 pathway. Figure 9 presented the effects of **F1** and **F2** on nuclear and cytosol AP-1 proteins level. After induction with LPS, the expressions of the c-Jun protein in the nucleus or cytoplasm were significantly increased. However, compared with the induction group, the expressions of the c-Jun protein in the nucleus were decreased by 47%, 58%, and 64% after 40, 60, and 80 μM treatment of **F1**, respectively. The same result was observed in the cytoplasm, and c-Jun expression in the cytoplasm was also significantly reduced; it reduced by 38%, 47%, and 51% after 40, 60, and 80 μM treatment of **F2**, respectively. The results are consistent, all of which are that the anti-inflammatory effect of **F1** is better than **F2**.

AP-1 of macrophages would be activated by LPS-induced. c-Jun in the AP-1 dimer increases and is involved in the expression of various inflammatory cytokines. When macrophages, in which c-Jun has been deleted induced by LPS, the mRNA expression of COX-2 was significantly lower than that of normal [20]. Therefore, it is speculated the COX-2 protein was inhibited by isoburmanol (**F1**) and burmanol (**F2**) through the retarding the expression of c-Jun protein; however, the two reference compounds, curcumin, and lucidone, could not significantly inhibit the c-Jun protein; therefore, they were unable to suppress COX-2 protein expression.

#### 2.3.4. Effects of Isoburmanol and Burmanol on MAPK

It is known that the mitogen-activated protein kinase (MAPK) family has the function of regulating AP-1; thus, the expression of MAPK proteins was further studied in this study. As shown in Figure 10, the expression of phosphorylated ERK, JNK, and p38 were all increased in macrophages by LPS induction. However, neither isoburmanol (**F1**) nor burmanol (**F2**) could reduce the expression of phosphorylated proteins. Therefore, it is inferred that the anti-inflammatory activity of **F1** and **F2** are mediated by directly reducing the avtivity of c-Jun to inhibit downstream iNOS and COX-2 proteins.

### 2.4. Molecular Docking Simulation Analysis

The isoburmanol (**F1**), burmanol (**F2**), curcumin, and ethyl p-coumarate were used to analyze the interaction between the target protein and the compound structure using molecular docking software (iGEMDOCK). AP-1, DNA-binding AP-1, JNK1, IKKβ, NF-κB, and these proteins can be classified into two inflammatory pathways (i.e., AP-1 and NF-κB pathways). If the binding energy is low, it means that the compound is more likely to interact with the protein. According to the binding energy of the protein and the compound, it can be divided into three grades, namely strong binding (binding energy < −100), moderate binding (binding energy between −100~−90), and weak binding (binding energy > −90). As shown in Table 5, curcumin possessed a strong binding activity to proteins in the NF-κB or AP-1 pathway, especially to proteins in the NF-κB pathway. This result is consistent with previous studies, curcumin also has a lower IC_50_ [21,22]. Conversely, ethyl p-coumarate binds weakly to proteins in the NF-κB or AP-1 pathways, and therefore has a high IC_50_.

Comparing the interaction of **F1** and **F2** with proteins in the two inflammatory pathways, these two compounds have a strong binding ability to proteins in the AP-1 pathway, but a weaker binding ability to proteins in the NF-κB pathway (Table 5). This is the reason that **F1** and **F2** could inhibit the protein expression of AP-1 but could not prevent the transfer of NF-κB into the nucleus. It is also consistent with the protein expression by Western blotting analysis (Figure 8 and Figure 9). Moreover, **F1** has a strong binding activity to AP-1 in the original state, while **F2** has only moderate binding ability. This result is also consistent with the IC_50_ result (**F1** < **F2**).

It has been reported that phosphorylated c-Jun is very unstable and will be degraded quickly, so it must form a dimer and bind to DNA as soon as possible to increase the gene expression of c-Jun, and then supplement c-Jun protein to form homeostasis [23]. However, the structure of AP-1 is a redox form. When AP-1 is in an oxidized state, it cannot bind to DNA. Only AP-1 in its original state can bind with DNA [24]. From the simulation results (Table 5), it can be found that **F1** has a strong binding ability to the original AP-1, a moderate binding ability to the oxidized AP-1, and a weak binding ability to the DNA-bound AP-1. The binding ability of **F2** also has the same trend. Therefore, combined with the results of the Western blotting analysis (Figure 9, the expression of c-Jun protein decreased), it is speculated that **F1** and **F2** should inhibit the binding of AP-1 to DNA, follow by decreased the activity of c-Jun.

The molecular structures of **F1** and **F2** are only the exchange of hydroxyl and methoxy groups at positions 9 and 10 (Figure 3), but they exhibited different anti-inflammatory activities. Therefore, the molecular docking simulation analysis was conducted to study the reasons for the different anti-inflammation activities of **F1** and **F2**. The interaction with AP-1 is shown in Figure 11. **F1** interacts with amino acids, including Cys172, Arg173, Arg177, Thr180, Cys285, Arg288, Lys289, and Arg292 on AP-1 through a van der Waals force. **F2** would interact with amino acids such as Cys172, Arg173, Arg177, Thr180, Arg288, Lys289, Arg292, and Arg295. Although **F2** and Arg173 could form a hydrogen bond, **F1** could form more van der Waals forces with the surrounding amino acids, which results in **F1** binding to AP-1 better than **F2** overall.

Based on the results obtained in this study, isoburmanol (**F1**) and burmanol (**F2**) would interact with AP-1, inhibit the binding of AP-1 to DNA, and cause AP-1 to fail to transcribe the related factors of inflammation. Since the binding ability of AP-1 and **F1** is stronger than that of F2, the inhibition activities of **F1** on downstream proteins and inflammatory hormones were stronger than **F2**.

## 3. Materials and Methods

### 3.1. General Experimental Procedures

Both 1D and 2D NMR experiments (^1^H, ^13^C, COSY, HSQC, and HMBC) were performed at 300 K in CD_3_OD on a Bruker AVANCEIII 400 spectrometer (Bruker, Billerica, MA, US). ESIMS were obtained in the negative ion mode on a Bruker amazon speed mass spectrometer (Bruker). Column chromatography was performed over silica gel (40–63 µm; Merck, Darmstadt, Germany). NP-HPLC separations were conducted on an Agilent 1100 series system equipped with a Luna 5 µm Silica (2) column (250 mm × 10 mm; Phenomenex). TLC analyses were carried out using Silica gel 60 F254 (Merck), and compounds were detected by sulfuric acid (Sigma-Aldrich, St. Louis, MO, USA).

### 3.2. Plant Material

The leaves of *C. insularimontanum* were collected in March 2020 in Aowanda National Forest Recreation Area and identified by Professor Yen-Hsueh Tseng (National Chung Hsin University). A voucher specimen (Y.S. Tseng 3125, TCF) has been deposited at the herbarium of the same university.

### 3.3. Preparation of C. insularimontanum Leaves Essential Oil (CILE) and GC/MS Analysis

Air-dried leaves were subjected to hydro-distillation in a Clevenger-type apparatus for 6 h, followed by the determination of oil contents. The CILE was stored in airtight sample vials prior to analysis of gas chromatography–mass spectrometry (GC–MS) and bioactivity evaluation. The ITQ 900 mass spectrometer coupled with a DB-5MS column was conducted to analyze the composition of CILE, and the temperature program was as follows: 40 °C for 3 min, then increased to 3 °C/min to 180 °C, and then increased to 20 °C /min to 280 °C hold for 5 min. The other parameters were injection temperature, 240 °C; ion source temperature, 200 °C; EI, 70 eV; carrier gas, He 1 mL/min; and mass scan range, 40–600 *m*/*z*. The volatile compounds were identified by Wiley/NBS Registry of mass spectral databases (V. 8.0, Hoboken, NJ, USA), National Institute of Standards and Technology (NIST) Ver. 2.0 GC/MS libraries and the Kovats indices were calculated for all volatile constituents using a homologous series of *n*-alkanes C9-C24. The major components were identified by co-injection with standards (wherever possible).

### 3.4. Extraction and Isolation

The fresh leaves of *C. insularimontanum* (4.9 kg) were extracted for 7 days with 70% EtOH to give 297.5 g crude. The crude was partitioned with solvents of increasing polarity: *n*-hexane, ethyl acetate, 1-butanol, and water to give Hex (34.2 g), EA (54.0 g), BuOH (83.4 g), and water (119.3 g) soluble fractions. The EA soluble fraction was subjected to silica gel column chromatography by eluting with *n*-hexane followed by increasing concentrations of ethyl acetate in *n*-hexane (between 5% and 100%) and used thin layer chromatography to afford 10 subfractions (A–J). Fraction F was subjected to normal phase HPLC on a Luna 5 μm Silica (2) column with *n*-hexane-ethyl acetate (1:1) as eluent to collect compounds **F1** and **F2**. Compound **F1**: white amorphous powder; ESIMS *m/z* 283.0 [M-H]^−^; ^1^H and ^13^C NMR data, see Table 1. Compound **F2**: white amorphous powder; ESIMS *m/z* 283.1 [M-H]^−^; for ^1^H and ^13^C NMR data, see Table 2.

### 3.5. Cell Culture and Cell Survival Assay

The monocyte/macrophage cell line (RAW 264.7) was obtained from the American Type Culture Collection (ATCC). Cells were maintained in Dulbecco’s Modified Eagle medium (DMEM; HyClone, Washington, DC, US), supplemented with 10% fetal bovine sera (FBS; Gibco, Waltham, MA, US), 1 mM sodium pyruvate (Corning), 1% glutamax (Gibco), and 1% penicillin/streptomycin (Corning) at 37 °C in a humidified atmosphere of 5% CO_2_. Cells were subcultured at a density of 1 × 10^6^ cells/mL after growing 85%. Stock solutions of samples in dimethyl sulfoxide (DMSO; Sigma-Aldrich) were stored in the dark at 4 °C. Appropriate dilutions were prepared on the day of the experiments. The final concentration of DMSO did not exceed 0.1% (*v*/*v*). Cell viability was measured using a colorimetric MTT assay. Cells were seeded in 96-well plates at a density of 1 × 10^5^ cells/well. After 24 h incubated with samples, the culture medium was removed and replaced with fresh medium containing 0.5 mg/mL of MTT (Sigma-Aldrich) for 1 h. The supernatant was removed and added 100 µL DMSO to dissolve formazan. Absorption at 570 nm for each well was measured using an ELISA microplate reader (BioTeck Instruments, Winooski, VT, US).

### 3.6. Anti-Inflammatory Assay

Anti-inflammatory activity was measured by nitric oxide (NO) and prostaglandin E_2_ (PGE_2_) productions. RAW 264.7 cells were seeded in 96-well plates at a density of 1 × 10^5^ cells/well. Adhered cells were then incubated for 24 h with or without 1 μg/mL of lipopolysaccharide (LPS; Sigma-Aldrich) in the absence or presence of samples. The supernatants detected were NO and PGE_2_. For the product of NO, the supernatants were mixed with an equal volume of Griess reagent (Sigma-Aldrich) and incubated for 15 min at room temperature. The nitrite concentration was measured at 540 nm using an ELISA microplate reader. PGE_2_ production in supernatants was measured using an ELISA Kit (Cayman Chemical, Ann Arbor, MI, USA) according to the manufacturer’s instructions.

### 3.7. Protein Extraction and Immunoblotting

RAW 264.7 cells were incubated with samples and 1 μg/mL of LPS for 18 h to detect iNOS and COX-2 expression, for 16 h to AP-1 expression, and for 1 h to MAPK expression. The cells were harvested and washed with ice-cold phosphate-buffered saline (PBS) twice and centrifuged at 2500× *g* for 5 min at 4 °C. The total protein was extracted with lysis buffer (Thermo Fisher Scientific, Waltham, MA, USA) containing a 1% protease inhibitor cocktail and 1% phosphatase protease inhibitor. Cytosolic and nuclear proteins were extracted using a nuclear extraction kit (Thermo Scientific) following the manufacturer’s protocol. The concentration of protein was determined by the Bradford protein assay (Bio-Rad, Hercules, CA, USA) according to the manufacturer’s instructions. Briefly, 80 μg of total protein or 25 μg of cytosolic and nuclear proteins were separated by 10% sodium dodecyl sulfate-polyacrylamide gel electrophoresis (SDS-PAGE) and transferred onto polyvinylidene difluoride (PVDF; Merck Millipore, Burlington, MA, USA) membranes at 300 mA for 120 min. The membranes were blocked with blocking buffer (5% w/v skim milk in TBST) for 1 h at room temperature and then incubated with primary antibody overnight at 4 °C, including anti-iNOS (dilution 1:1000; 160862) from Cayman Chemical Company, anti-β-actin (dilution 1:500; Sc-47778) from Santa Cruz Biotechnology, Inc., anti-GAPDH (dilution 1:500; ab8245), anti-NF-κB p105/p50 (dilution 1:400; ab7971) from Abcam plc., and anti-Cox2 (dilution 1:1000; #4842), anti-NF-κB p65 (dilution 1:1000; #8242), anti-c-Jun (dilution 1:1000; #9165), anti-JNK (dilution 1:1000; #9258), anti-phospho-JNK (Thr183/Tyr185; dilution 1:1000; #9255), anti-p38 (dilution 1:1000; #9212), anti-phospho-p38 (Thr180/Tyr182; dilution 1:1000; #9211), anti-ERK1/2 (dilution 1:1000; #9102), and anti-phospho-ERK1/2 (dilution 1:1000; #4377) from Cell Signaling Technology, Inc. Next, after washing with TBST three times, they were incubated with secondary antibodies (anti-rabbit or anti-mouse conjugated with horseradish peroxidase) for 2 h. Protein bands were detected by immobilon western chemiluminescent HRP substrate (Merck Millipore), visualized using ChemiDoc XRS+ docking system (Bio-Rad), and quantified by Imagelab software (Bio-Rad).

### 3.8. Molecular Docking

The iGEMDOCK was used to simulate the compound of isoburmanol, burmanol, curcumin, and ethyl *p*-coumarate into the active site of the protein, including AP-1 (PDB code: 5VPA), AP-1 of the oxidized form (PDB code: 5VPB), AP-1 of bound to DNA (PDB code: 1FOS), JNK1 (PDB code: 3V3V), IKKβ (PDB code:4KIK), and NF-κB (PDB code: 1BFT) to calculate the binding energy of the compound-protein complexes. In silica docking was predicted for each compound-protein complex. The best binding orientation between compound and protein at the molecular docking stage needs to minimize the energy when the compound docking process to the protein. PyMOL and LigPlot 2.2.4 was used for visualizing the docking three-dimensional (3D) and two-dimensional (2D) conformations.

## 4. Conclusions

This study confirmed that the essential oil of *C. insularimontanum* leaves has good anti-inflammatory activity, and IC_50_ was 28.8 μg/mL in the LPS-stimulated RAW 264.7 cell assay. The 1,8-cineole (35.965) was the most abundant constituent of essential oil, and 1,8-cineole has been proven to have good anti-inflammatory activity [25]. Therefore, the essential oil of *C. insularimontanum* leaves with 1,8-cineole as the main component has great potential as an anti-inflammatory drug. In addition to the essential oil, the IC_50_ value for *C. insularimontanum* ethanolic extract was 36.7 μg, it also exhibits excellent anti-inflammatory activity. Two compounds, isoburmanol (**F1**) and burmanol (**F2**), were obtained by bioactivity isolation protocol from the ethanolic extracts. The IC_50_ for NO inhibition of **F1** and **F2** was 14.2 ± 3.6 μM and 44.7 ± 8.4 μM, respectively, and both compounds could also inhibit the PGE2 production. Although both **F1** and **F2** possessed a significant anti-inflammation activity, the activity of **F1** was better than **F2**. Through molecular docking simulation analysis, the results show that **F1** and **F2** would interact with AP-1, inhibit the binding of AP-1 to DNA, and cause AP-1 to fail to transcribe related factors of inflammation. The binding ability of AP-1 and **F1** was stronger than **F2**, and that is the reason why **F1** exhibited better activities in both downstream proteins and inflammatory cytokines. The anti-inflammation activity of **F1** and **F2** was different from the common pathway of inhibiting NF-κB. Both of them could inhibit the production of NO and PGE2 by directly inhibiting AP-1 (c-Jun) protein and then inhibiting the downstream iNOS and COX-2 (Figure 12). The above test results show that both the essential oil and **F1** and **F2** isolated from ethanolic extracts have good anti-inflammatory activity, and they are expected to be used as a reference for the development of medical care products in the future.

## Figures and Tables

**Figure 1 plants-11-03252-f001:**
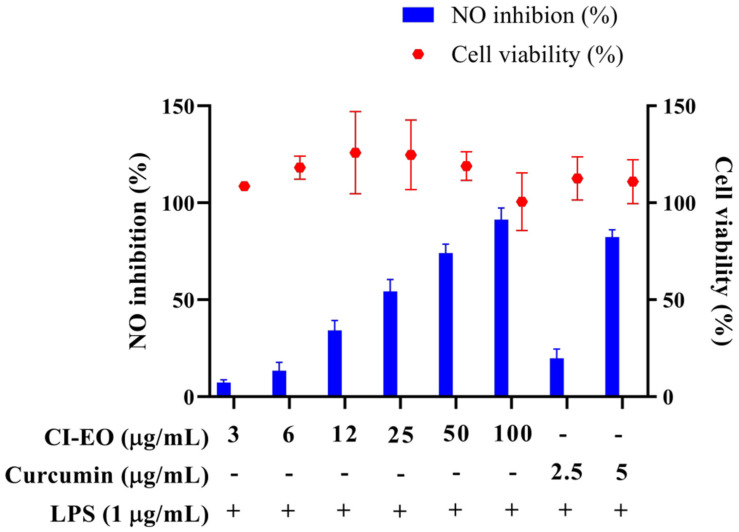
The effect of *C. insularimontanum* leaves’ essential oil on cell viability and NO inhibition. RAW 264.7 were treated with essential oil (3, 6, 12, 25, 50, and 100 μg/mL) and curcumin (2.5 and 5 μg/mL) with or without 1 μg/mL LPS for 24 h. The cell viability was analysed by the MTT method, NO inhibition was analysed by Griess assay. The data are represented as mean ± S.D. (n ≧ 3).

**Figure 2 plants-11-03252-f002:**
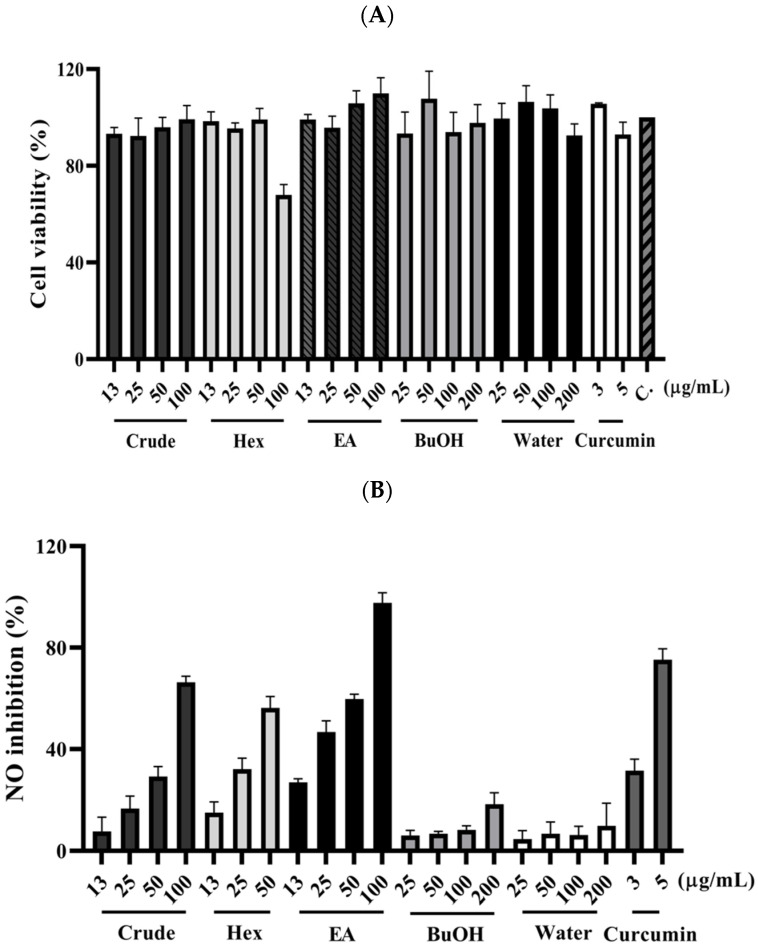
The effect of *C. insularimontanum* leaves’ ethanol extracts and its soluble fraction on cell viability (**A**) and NO inhibition (**B**). RAW 264.7 were treated with ethanol extracts and EA soluble fraction (13, 25, 50, and 100 μg/mL), *n*-Hex soluble fraction (13, 25, and 50 μg/mL), BuOH, and water-soluble fraction (25, 50, 100, and 200 μg/mL), and curcumin (3 and 5 μg/mL) with 1 μg/mL LPS for 24 h. The NO inhibition was analysed by Griess assay. The data are represented as mean ± S.D. (n ≧ 3).

**Figure 3 plants-11-03252-f003:**
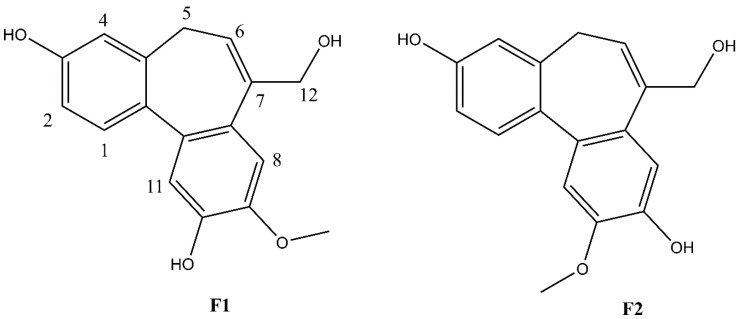
Structures of **F1** and **F2**.

**Figure 4 plants-11-03252-f004:**
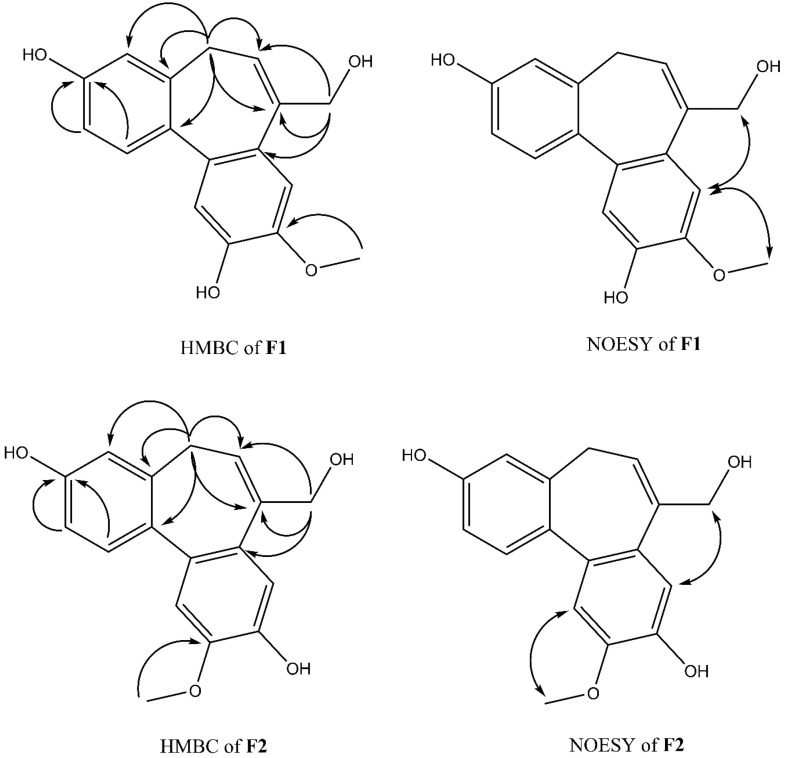
Key HMBC and NOESY correlations of **F1** and **F2**.

**Figure 5 plants-11-03252-f005:**
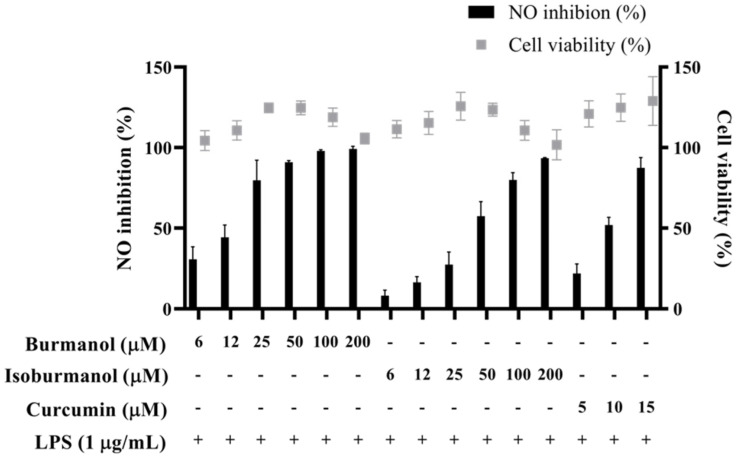
The effects of isoburmanol (**F1**) and burmanol (**F2**) on cell viability and NO inhibition. RAW 264.7 were treated with isoburmanol and burmanol (6, 12, 25, 50, 100, and 200 μM) and curcumin (5, 10, and 15 μM) with 1 μg/mL LPS for 24 h. The cell viability was analysed by MTT method, and NO inhibition was analysed by Griess assay. The data are represented as mean ± S.D. (n ≧ 3).

**Figure 6 plants-11-03252-f006:**
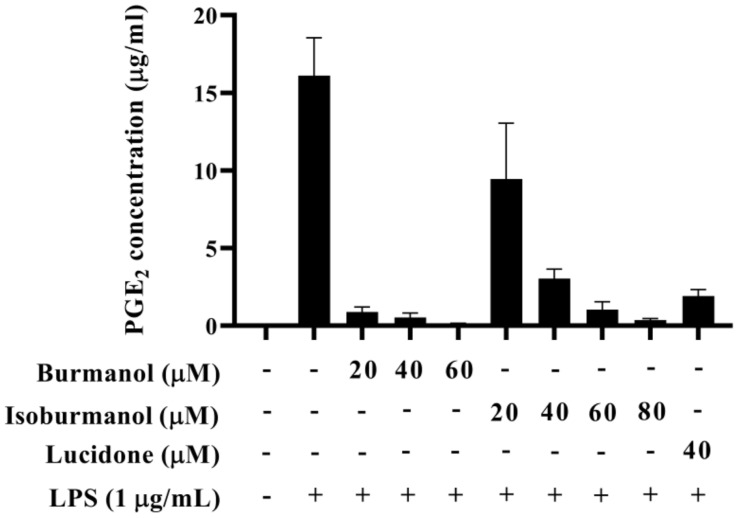
The effect of isoburmanol (**F1**) and burmanol (**F2**) on PGE_2_ concentration. RAW 264.7 were treated with isoburmanol (20, 40, and 60 μM) and burmanol (20, 40, 60, and 80 μM), and lucidone (40 μM) with 1 μg/mL LS for 24 h. The PGE_2_ concentration was analysed by PGE_2_ ELISA kit. The data are represented as mean ± S.D. (n ≧ 3).

**Figure 7 plants-11-03252-f007:**
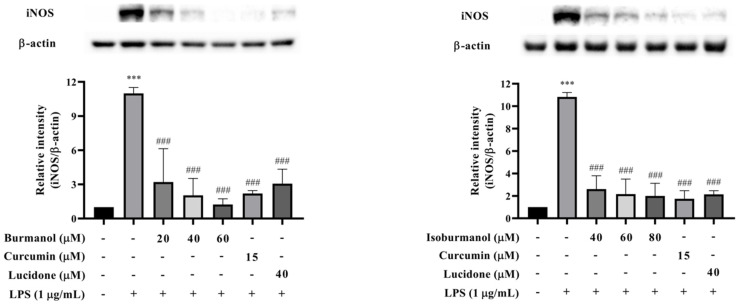
The effects of isoburmanol (**F1**) (**left**) and burmanol (**F2**) (**right**) on iNOS protein levels. RAW 264.7 were treated with **F1** (20, 40, and 60 μM), **F2** (40, 60, and 80 μM), curcumin (15 μM) and lucidone (40 μM) with 1 μg/mL LPS and collected protein after 18 h. Bar charts are the quantized results of Western blot. The data are represented as mean ± S.D. (n ≧ 3). Statistical analysis using one-way ANOVA. *** *p* < 0.001, compared with the vehicle group. ### *p* < 0.001, compared with the only LPS-induced group.

**Figure 8 plants-11-03252-f008:**
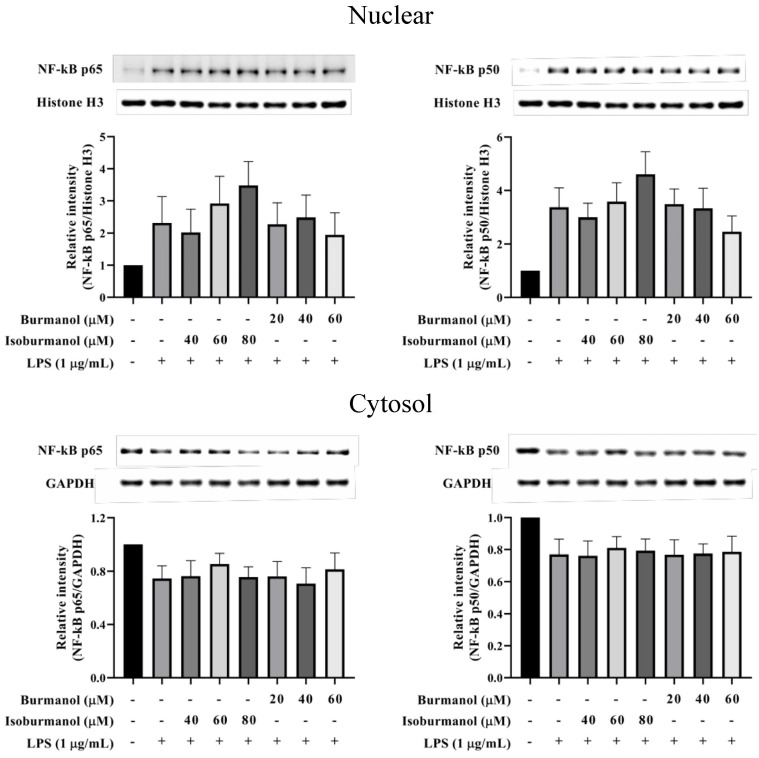
The effect of isoburmanol (**F1**) and burmanol (**F2**) on nuclear (upper) and cytosol (down) NF-κB protein levels. RAW 264.7 were treated with **F1** (20, 40, and 60 μM) and **F2** (40, 60, and 80 μM) with or without 1 μg/mL LPS and collected protein after 1 h. Bar charts are the quantized results of western blot. The data are represented as mean ± S.D. (n ≧ 3).

**Figure 9 plants-11-03252-f009:**
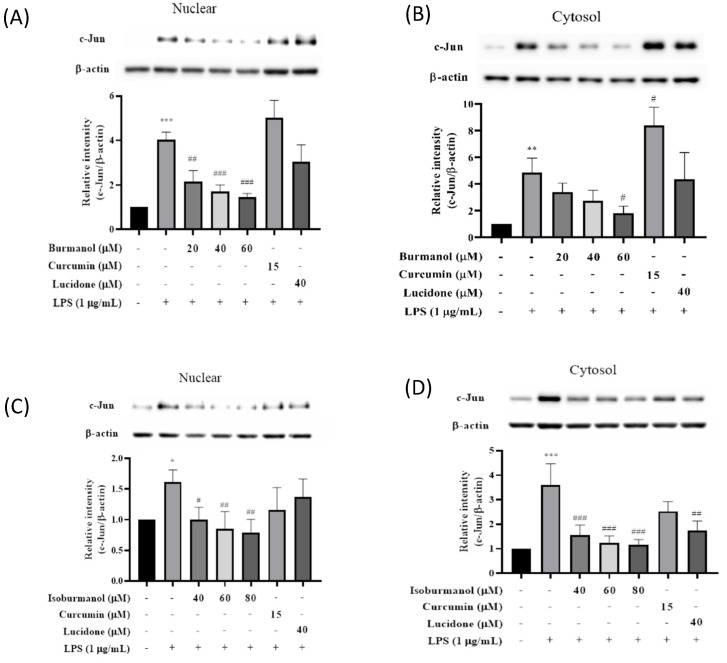
The effect of (**A**) isoburmanol (**F1**) on nuclear, (**B**) isoburmanol (**F1**) on cytosol, (**C**) burmanol (**F2**) on nuclear and (**D**) burmanol (**F2**) on cytosol AP-1 protein level. RAW 264.7 were treated with isoburmanol (20, 40, and 60 μM), burmanol (40, 60, and 80 μM), curcumin (15 μM) and lucidone (40 μM) with or without 1 μg/mL LPS, and collected protein after 16 h. Bar charts are the quantized results of Western blot. The data are represented as mean ± S.D. (n ≧ 3). Statistical analysis using one-way ANOVA. * *p* < 0.05, ** *p* < 0.01 and *** *p* < 0.001, compared with the vehicle group. # *p* < 0.05, ## *p* < 0.01 and ### *p* < 0.001, compared with LPS induced group.

**Figure 10 plants-11-03252-f010:**
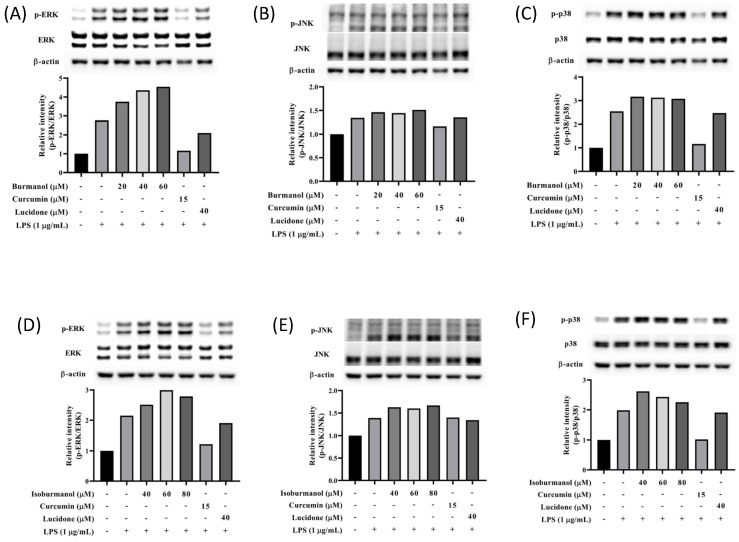
The effects of isoburmanol (**F1**) on phosphorylation of MAPK protein ((**A**), ERK; (**B**), JNK; (**C**), p38) level. RAW 264.7 were treated with **F1** (20, 40, and 60 μM), curcumin (15 μM), and lucidone (40 μM) with or without 1 μg/mL LPS and collected protein after 1 h. Bar charts are the quantized results of Western blot. The effect of burmanol on phosphorylation of MAPK ((**D**), ERK; (**E**), JNK; (**F**), p38) protein level. RAW 264.7 were treated with burmanol (40, 60, and 80 μM), curcumin (15 μM), and lucidone (40 μM) with or without 1 μg/mL LPS and collected protein after 1 h. Bar charts are the quantized results of Western blot.

**Figure 11 plants-11-03252-f011:**
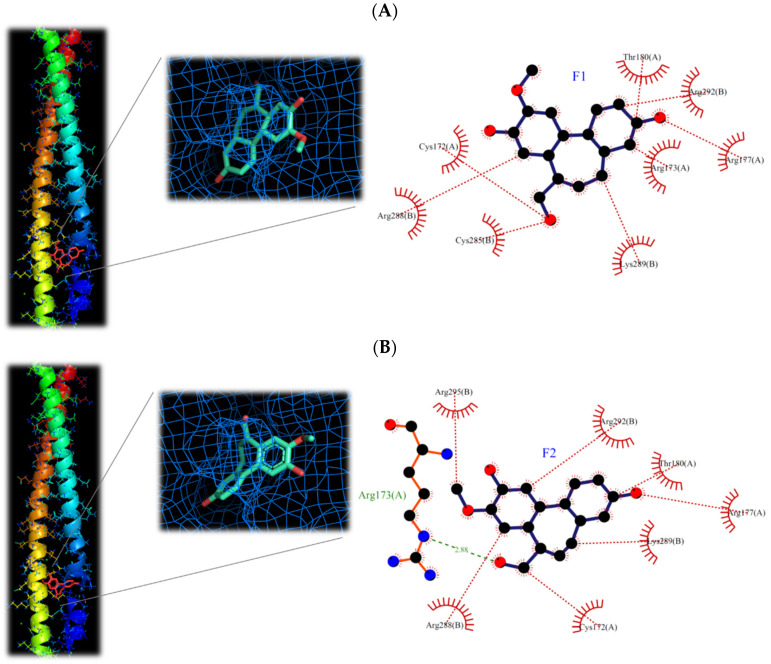
3D and 2D structures of interaction between (**A**) isoburmanol (**F1**) and (**B**) Burmanol (**F2**) with AP-1 (5VPA).

**Figure 12 plants-11-03252-f012:**
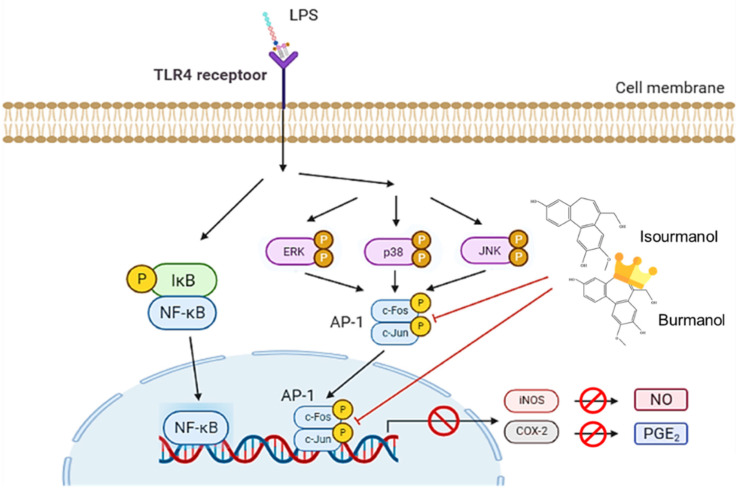
The anti-inflammatory mechanism of isoburmanol (**F1**) and burmanol (**F2**) in RAW 264.7.

**Table 1 plants-11-03252-t001:** Chemical composition of the essential oils (EO) from *C. insularimontanum* leaves.

RT (min)	Name	Concentration%	KI ^a^	Identification ^b^
15.97	α-Pinene	7.55	934	KI/MS/ST
17.12	Camphene	2.44	950	KI/MS/ST
18.97	Sabinene	5.06	974	KI/MS/ST
19.12	β-Pinene	3.14	976	KI/MS/ST
21.93	α-Terpinene	0.53	1014	KI/MS/ST
22.44	m-Cymene	0.74	1023	KI/MS
22.69	Limonene	4.14	1028	KI/MS/ST
22.85	1,8-Cineole	35.94	1030	KI/MS/ST
24.51	γ-Terpinene	2.46	1059	KI/MS/ST
31.21	4-Terpeneol	1.54	1180	KI/MS/ST
32.02	α-Terpineol	3.61	1194	KI/MS/ST
36.29	Isobornyl acetate	4.81	1283	KI/MS/ST
40.27	α-Cubebene	0.46	1371	KI/MS
42.13	Caryophyllene	0.93	1413	KI/MS/ST
45.06	(+)-Valencene	0.85	1483	KI/MS
45.36	β-Guaiene	0.37	1490	KI/MS
46.32	Cubebol	0.61	1514	KI/MS/ST
47.21	α-Calacorene	0.55	1537	KI/MS
47.57	β-Elemol	0.93	1546	KI/MS/ST
48.77	Nerolidol	0.87	1576	KI/MS/ST
50.79	Guaiol	0.83	1628	KI/MS
51.61	α-Eudesmol	6.17	1650	KI/MS/ST
64.95	Kaurene	1.47	2034	KI/MS/ST

^a^ Kovats index on DB-5MS column in reference to n-alkanes. ^b^ MS, NIST library and literature; KI, Kovats index; ST, authentic standard compounds.

**Table 2 plants-11-03252-t002:** Yield and NO inhibition activity of sub-fraction derived from ethyl acetate soluble extracts of *C. insularimontanum* leaves’ ethanol extracts.

Fractions	A	B	C	D	E	F	G	H	I	J	CUR ^a^
Weight (g)	0.1	0.3	0.7	1.5	5.0	4.2	2.4	3.2	1.8	3.4	-
IC_50_ ^b^ (μg/mL)	>40	>20	19.0	25.5	37.2	15.8	23.5	25.1	33.9	38.0	2.6

^a^ CUR: curcumin. ^b^ IC_50_: half-maximal inhibitory concentration.

**Table 3 plants-11-03252-t003:** ^1^H and ^13^C NMR chemical shift data of **F1** (isoburmanol) (*δ* ppm, in CD_3_OD).

Positions	^13^C	^1^H	HMBC
1	131.5	7.26 (d, *J* = 8.3 Hz, 1H)	H-1/C-2, H-1/C-3, H-1/C-11a, H-1/C-11b
2	114.1	6.70 (dd, *J* = 8.3, 2.5 Hz, 1H)	H2/C-1, H-2/C-3, H-2/C-4
3	158.1		
4	114.0	6.65 (d, *J* = 2.5 Hz, 1H)	H-4/C-2, H-4/C-3, H-4/C-4a, H-4/C-5
4a	131.8		
5	34.1	Ha: 2.72 (dd, *J* = 12.6, 6.3 Hz, 1H)	H-5/C-4, H-5/C-4a, H-5/C-6, H-5/C-7, H-5/C-11b
		Hb: 3.03 (dd, *J* = 12.6, 8.2 Hz, 1H)	
6	128.0	6.13 (t, *J* = 7.4 Hz, 1H)	H-6/C-5, H-6/C-7a, H-6/C-12
7	139.1		
7a	129.6		
8	110.8	7.18 (s, 1H)	H-8/C-7, H-8/C-9, H-8/C-10
9	147.8		
10	146.5		
11	117.2	7.03 (s, 1H)	H-11/C-7a, H-11/C-9, H-11/C-11a
11a	135.6		
11b	145.1		
12	66.4	Ha: 4.33 (d, *J* = 13.0 Hz, 1H)	H-12/C-6, H-12/C-7, H-12/C-7a
		Hb: 4.45 (d, *J* = 13.0 Hz, 1H)	
OMe	56.6	3.92 (s, 3H)	OMe/C-9

s: singlet, d: doublet, dd: doublet of doublets, t: triplet.

**Table 4 plants-11-03252-t004:** ^1^H and ^13^C NMR chemical shift data of **F2** (burmanol) (*δ* ppm, in CD_3_OD).

Positions	^13^C	^1^H	HMBC
1	131.4	7.32 (d, *J* = 8.4 Hz, 1H)	H-1/C-3, H-1/C-11a, H-1/C-11b
2	114.3	6.71 (dd, *J* = 8.4, 2.6 Hz, 1H)	H-2/C-3, H-2/C-4
3	158.3		
4	114.1	6.64 (d, *J* = 2.6 Hz, 1H)	H-4/C-2, H-4/C-3, H-4/C-4a, H-4/C-5
4a	132.0		
5	34.1	Ha: 2.72 (dd, *J* = 12.6, 6.3 Hz, 1H)	H-5/C-4, H-5/C-4a, H-5/C-6, H-5/C-7, H-5/C-11b
		Hb: 3.03 (dd, *J* = 12.6, 8.2 Hz, 1H)	
6	127.9	6.13 (t, *J* = 7.6 Hz, 1H)	H-6/C-5, H-6/C-7, H-6/C-12
7	138.9		
7a	130.8		
8	114.1	7.06 (s, 1H)	H-8/C-7, H-8/C-9, H-8/C-10
9	146.5		
10	148.1		
11	113.7	7.08 (s, 1H)	H-11/C-7a, H-11/C-9, H-11/C-10, H-11/C-11a
11a	134.2		
11b	145.2		
12	66.1	Ha: 4.30 (d, *J* = 13.0 Hz, 1H)	H-12/C-6, H-12/C-7, H-12/C-7a
		Hb: 4.39 (d, *J* = 13.0 Hz, 1H)	
OMe	56.5	3.93 (s, 3H)	OMe/C-10

s: singlet, d: doublet, dd: doublet of doublets, t: triplet.

**Table 5 plants-11-03252-t005:** Using molecular docking to simulate the interaction between compound and target protein.

Compound	Protein	PDB Code	IC_50_ (μM)	Minimum Binding Energy
Isoburmanol	AP-1	5VPA	14.0	−103.53
	AP-1 (oxidized form)	5VPB		−95.50
	AP-1 (bound to DNA)	1FOS		−85.33
	JNK1 (MYU binding sites)	3V3V		−96.61
	IKKβ (KSA binding sites)	4KIK		−94.71
	NF-κB	1BFT		−86.78
Burmanol	AP-1	5VPA	43.8	−98.88
	AP-1 (oxidized form)	5VPB		−87.82
	AP-1 (bound to DNA)	1FOS		−88.51
	JNK1 (MYU binding sites)	3V3V		−91.95
	IKKβ (KSA binding sites)	4KIK		−90.82
	NF-κB	1BFT		−89.70
Curcumin	AP-1	5VPA	9.4	−113.54
	AP-1 (oxidized form)	5VPB		−113.95
	AP-1 (bound to DNA)	1FOS		−103.62
	JNK1 (MYU binding sites)	3V3V		−114.44
	IKKβ (KSA binding sites)	4KIK		−122.01
	NF-κB	1BFT		−120.81
Ethyl p-coumarate	AP-1	5VPA	58.0	−89.43
	AP-1 (oxidized form)	5VPB		−82.96
	AP-1 (bound to DNA	1FOS		−72.42
	JNK1 (MYU binding sites)	3V3V		−77.13
	IKKβ (KSA binding sites)	4KIK		−73.09
	NF-κB	1BFT		−74.87

## Data Availability

Not applicable.

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
