# Peer review of "Anti-Inflammatory Activities of Constituents from Cinnamomum insularimontanum Hayata Leaves and Their Mechanisms"

_plants, 2022, doi:10.3390/plants11233252_

Round 1

Reviewer 1 Report

Overview and general recommendation

Thanks for the opportunity to review this research. In relation to the manuscript titled " Anti-inflammatory Principals of Cinnamomum insularimontanum Hayata Leaves and their mechanisms ", the manuscript in its present form is not well prepared as there is serious discord with the flow of information. The methodology is not well prepared, and it would be lame for the authors to expect the audience to understand such good results when some aspects are suddenly brought into the discussion without a prior explanation in the methodology section. In other words, the discussion is not easily comprehensible without adequate description in the methodology section. Furthermore, the discussion lacks elaboration on the scientific merit of the results and their contribution to the scientific community in general and the field. However, below are some recommendations to restructure and improve the manuscript for resubmission.

Abstract

The aim is not aligned with the background provided. The methodology is not well described in the abstract.

Introduction

The introduction is not focused and needs additional information. The authors should also highlight the importance of the study.

Results and discussion

The discussion will be difficult to follow if the authors do not explain the different treatments from the methodology section as well as the objective of each treatment.

I suggest that the authors restructure the charts with consistent legends on the chart.

All parameters presented in the results section must be aligned with sufficient background information in the introduction for the audience to make scientific sense of the presented information.

The scientific merit of the current work is not exposed appropriately in this manuscript and that makes the work sounds irrelevant although many experiments were conducted.

Conclusion

Given the current state of the manuscript, a more appropriate conclusion must be rewritten or aligned when the manuscript is improved.

There are some technical errors in the text and the references. Please check them. I recommend the publishing of the paper after the necessary corrections.

Author Response

Reviewer: 1

Overview and general recommendation: The methodology is not well prepared, and it would be lame for the authors to expect the audience to understand such good results when some aspects are suddenly brought into the discussion without a prior explanation in the methodology section. In other words, the discussion is not easily comprehensible without adequate description in the methodology section. Furthermore, the discussion lacks elaboration on the scientific merit of the results and their contribution to the scientific community in general and the field. However, below are some recommendations to restructure and improve the manuscript for resubmission.

Response:  We would appreciate the reviewer-1, whom provided a positive comment on our work. The reviewer’s comments and our responses are as follows.

Abstract: The aim is not aligned with the background provided. The methodology is not well described in the abstract.

Response: Thank you for pointing this out. As per the your suggests we modified the abstract, especially mention why the anti-inflammation activity was selected to study for the metabolites of Cinnamomum insularimontanum Hayata Leaves? Besides, the structure analysis method for compounds is also described in the revised version. As regard to NO, PGE2 inhibition assay as well as proteins expression analysis are common and well-known technologies and study strategy for investing anti-inflammation activity of natural products.   

Introduction: The introduction is not focused and needs additional information. The authors should also highlight the importance of the study.

Response:  Thank you for your kind reminding. As per the reviewer’s suggestion, we added the sentences to highlight our study. “Although we had demonstrated the anti-inflammation activity of C. insularimontanum fruits, the bioactivity investment for leaves is rare. In this study, the essential oil was obtained from C. insularimontanum leaves, and its compositions and anti-inflammation activity were characterized and evaluated. On the other hand, the anti-inflammatory principals in ethanolic extracts of C. insularimontanum Leaves were also obtained follow by bioactivity guided fractionation procedure. According to our finding, it is expected to be used extracts and compound isolated from C. insularimontanum leaves as a reference for the development of medical care products in the future”.

Results and discussion: The discussion will be difficult to follow if the authors do not explain the different treatments from the methodology section as well as the objective of each treatment.

Response:  Thank you for pointing this out. As we mentioned, the anti-inflammation study methods conducted in this study are common and well-known technologies and study strategy for investing anti-inflammation activity of natural products.  We approved the anti-inflammation activity on C. insularimontanum leaves. Furthermore, the anti-inflammation mechanism of bioactivity compounds, burmanol (F1) and isoburmanol (F2),  were different from the common pathway of inhibiting NF-κB. Both of them could inhibit the production of NO and PGE2 by directly inhibiting AP-1 (c-Jun) protein and then inhibiting the downstream iNOS and COX-2. Although both F1 and F2 possessed a significant anti-inflammation activity. Through molecular docking simulation analysis, the results show that F1 and F2 would interact with AP-1, inhibit the binding of AP-1 to DNA, and caused AP-1 to fail to transcribe related factors of inflammation. The binding ability of AP-1 and F1 was stronger than F2, that is the reason F1 exhibited better activities in both downstream proteins and inflammatory cytokines.We believe that is a important results of this study.

Reviewer 2 Report

Dear Authors,

Presented manuscript contains a lot of experiments, which were quite well described, but I found a few problems:

1.       In abstract and other parts you wrote that the extract contains 16.17% of alfa-eudesmol, but in Table 1 there is a value 6.17%. Which value is correct?

2.       In Abstract you wrote “The essential oil of C. insularimontanum leaves was extracted by water distillation” but in point 2.2. you wrote „The fresh leaves of C. insularimontanum were extracted with 70% ethanol (EtOH)”. What was the correct solvent?

3.       Please check the caption of Figure 1 – ”Inhibitory activity of tyrosinase from mushroom by wood essential oils of Calocedrus formosana”??? Something is wrong in this caption.

4.       Point 2.1. „Chemical composition analysis and its anti-inflammation activity of C. insularimontanum leaves essential oil (CILE)” – you wrote „The nitric oxide (NO) inhibitory activity of fruit essential oil of CILE was 106 evaluated by using a LPS-stimulated RAW 264.7 cell assay”. What extract did you analyse, extract form fruit or leaves?

5.       Figure 1 in line 201- it shoud be Figure 4!

6.       Figure 5 – in caption of this figure you wrote „The effect of burmanol (F1) and isoburmanol (F2) on PGE2 concentration. RAW 264.7 were treated with 208 burmanol and isoburmanol (6122550100, and 200 μM)” but in the legend of Figure 5 there are quite different concentrations of formulation F1 and F2. Please correct it.

7. Explain please, why did you use different concentrations of burmanol and  isoburmanol in your experiments (Figure 5, 6, 7 and 8)?

Author Response

Reviewer: 2

Reviewer’s opinion: Presented manuscript contains a lot of experiments, which were quite well described

Response:  We thank the reviewer 2 provided such a positive and constructive comments on our work.

Comment 1: In abstract and other parts you wrote that the extract contains 16.17% of alfa-eudesmol, but in Table 1 there is a value 6.17%. Which value is correct?

Response: We apology for our typo, the content of α-eudesmol is 6.17%. We apricate your kind reminding.

Comment 2: In Abstract you wrote “The essential oil of C. insularimontanum leaves was extracted by water distillation” but in point 2.2. you wrote „The fresh leaves of C. insularimontanum were extracted with 70% ethanol (EtOH)”. What was the correct solvent?

Response: Thank you for pointing this out. Yes, it is correct. There are two different extracts studied in this study, which were essential oil prepared by water distillation (discussed in 2.1) and methanol extracts (discussed in 2.2).

Comment. 3 Please check the caption of Figure 1 – ”Inhibitory activity of tyrosinase from mushroom by wood essential oils of Calocedrus formosana”??? Something is wrong in this caption.

Response: I'm so sorry we made an undeserved mistake. The figure legend of figure 1 is corrected intoThe effect of C. insularimontanum leaves essential oil on cell viability and NO inhibition. RAW 264.7 were treated with essential oil (3、6、12、25、50 and 100 μg/mL) and curcumin (2.5 and 5 μg/mL) with or without 1 μg/mL LPS for 24 hr. The cell viability was analysis by MTT method, NO inhibition was analysis by griess assay. The data are represented as mean ±S.D. (n≧3)”. Thanks for your kind reminding.

Comment 4: Point 2.1. „Chemical composition analysis and its anti-inflammation activity of C. insularimontanum leaves essential oil (CILE)” – you wrote „The nitric oxide (NO) inhibitory activity of fruit essential oil of CILE was 106 evaluated by using a LPS-stimulated RAW 264.7 cell assay”. What extract did you analyze, extract form fruit or leaves?

Response: Thank you for pointing this out. All the extracts, including essential oil analyzed in this study is extracted from leaves.

Comment 5: Figure 1 in line 201- it should be Figure 4!

Response: Yes, it is correct. Thank you for pointing this out.

Comment 6: Figure 5 – in caption of this figure you wrote „The effect of burmanol (F1) and isoburmanol (F2) on PGE2 concentration. RAW 264.7 were treated with burmanol and isoburmanol (6、12、25、50、100, and 200 μM)” but in the legend of Figure 5 there are quite different concentrations of formulation F1 and F2. Please correct it.

Response: Thanks for your correction, we revise the right the figure legend into “The effect of burmanol (F1) and isoburmanol (F2) on PGE2 concentration. RAW 264.7 were treated with burmanol (20, 40, and 60 μM) and isoburmanol (20, 40, 60, and 80 μM) and lucidone (40 μM) with or without 1 μg/mL LS for 24 hr.”

Comment 7 Explain please, why did you use different concentrations of burmanol and isoburmanol in your experiments (Figure 5, 6, 7 and 8)?

Response: The reason using the difference dosage for F1 and F2 in protein expression analysis is the anti-inflammation activity of F1 is stronger than F2. The NO and PGE2 production for F1 treatment was almost inhibited when dosage higher than 60 μM, thus we used the highest dosage of F1 was 60 μM.

Reviewer 3 Report

Dear authors,

Please find my comments on your article "Anti-inflammatory Principals of Cinnamomum insularimontanum Hayata Leaves and their mechanisms"

1.      Line 29 correct …was firstly isolated… to …it was isolated for the first time…

2.      Line 31 rephrase the sentence …respectively in RAW 264.7 cells induction of lipopolysaccharide… (…RAW 264.7 cells after induction of lipopolysaccharide or …RAW 264.7 cells stimulated by lipopolysaccharide…)

3.      Sentence in line 30/33 is unclear; …They also revealed the significant prostaglandin E2 (PGE2) production inhibition F1 and F2 could significantly inhibit iNOS and COX-2 proteins expression…

4.      Line 36 – it is more common to use anti-inflammatory than anti-inflammation activity

5.      Sentence in line 48 …Inflammation is the origin and progression of various diseases… could be rephrased to …Inflammation is on the basis of various diseases, such as cancer, diabetes, asthma, inflammatory bowel disease, multiple sclerosis, Alzheimer’s disease, and can contribute to their progression…

6.      Line 67: when you mention the plant species for the first time in text, please write the formal scientific plant name: Cinnamomum insularimontanum Hayata. According to the world flora (https://wfoplantlist.org/plant-list/taxon/wfo-0000605135-2022-06) the correct name is Cinnamomum insulari-montanum Hayata, but

7.      Line 85 –"following" instead follow

8.      Line 90 – it is more common to use anti-inflammatory instead anti-inflammation; delete "its" in subtitle 2.1

9.      Line 92 - the yield of hydro distillation is not correct, 2.1% is not equivalent to 2.1 mL/kg

10.   Line 93 - The relative contents (%) of major compounds in CFEO are (not is) shown

11.   Line 102 – Table. 1 - Please put the plant name in italic

12.   Line 109/110 – correct the sentence …The nitric oxide (NO) inhibitory activity of fruit essential oil of CILE was evaluated by using a LPS-stimulated RAW 264.7 cell assay… CILE is an acronym for essential oil from leaves so it is impossible to have activity of fruit essential oil of CILE

13.   Line 110 - should be Fig. 1 instead Fig. 2.

14.   Line 116– why is Zingiber cassumunar essential oil relevant for this study?

15.   Line 121 – In the Figure 1 description concentrations should be separated by commas: (3, 6, 12, 25, 50, and 100 µg/ml)

16.   Line 125, Title 2.2. it is more common to use anti-inflammatory instead anti-inflammation; please remove CILE from the title while in this chapter you have tested fresh leaves extract and not essential oil

17.   Line 129 it is more common to use anti-inflammatory instead anti-inflammation

18.   Line 131 – please put "cell model" instead "cells modele"

19.   Line 131 – Please add "EA soluble fraction presented a more potent anti-inflammatory activity…"

20.   Line 143/147 – tested concentrations should be separated by commas; please explain the fact that cells were treated with or without LPS (which ones were treated?); put the capital letter on Griess assay

21.   Line 151 – please write "derived" instead "derive"

22.   Line 185 –Table 3. 1H and 185 13C NMR chemical shift data of F1 (burmanol), but in previous text in line 175/175 F1 was named as isoburmanol

23.   Line 190 - Table 4. 1H and 190 13C NMR chemical shift data of F2 (isoburmanol), but in previous text in line 178 F2 was named as burmanol. Please check for further discrepancies in the work.

24.   Line 206 – The figure is wrongly numerated, it should by Figure 4 (not Figure 1)

25.   Line 207 – tested concentrations should be separated by commas

26.   Line 208 - please explain the fact that cells were treated with or without LPS since on the figure it is clearly stated that LPS was put in every tested solution

27.   Line 217 – in title 2.3.2. Effects of F1 and F2 on iNOS and COX-2 expression in LPS-induced macrophages it is suggested that Cox-2 expression was measured, but there is no supporting data or figure in the text

28.   Line 228/229 – please write burmanol (F1) and isoburmanol(F2) to be consistent with previous figures; tested concentrations should be separated by commas

29.   Line 232 – It is stated that F1 and F2 influence on cox-2 expression but there are no supporting results/figures.

30.   Line 245 - please write burmanol (F1) and isoburmanol (F2) to be consistent with previous figures

31.   Line 246 - tested concentrations should be separated by commas

32.   Line 253 – citation was inserted as the name of the author and year (Li et al., 2016) instead adding a number in brackets

33.   Line 270 – to be consistent with previous figure descriptions please add F1 and F2 and separate tested concentrations by commas

34.   Line 282 – please correct the sentence …However, neither burmanol (F1) nor isoburmanol (F2) could not reduce… to …However, neither burmanol (F1) nor isoburmanol (F2) could reduce

35.   Line 291 – please write isoburmanol (F2); separate tested concentrations by commas

36.   Line 459-461 – please rephrase the sentence "In addition to the essential oil, the ethanolic extract of the NO inhibition rate IC50 was 36.7 μg also exhibits excellent anti-inflammatory activity, the IC50 for NO inhibition was 36.7 μg/ml"

37.   Line 463 – The IC50 values are not the same as previously mentioned in line 195

38.   Line 487-546 – please correct the references. In the Instruction for authors, it is stated that journals should be listed as abbreviations, not as the full journal name 

Author Response

Please see in the attachment.
